

# Worldwide lake level trends and responses to background climate variation

**Benjamin M. Kraemer[1,3], Anton Seimon[2], Rita Adrian[1], Peter B. McIntyre[3,4]**

[1]Ecosystem Research Department, IGB Leibniz Institute of Freshwater Ecology and Inland Fisheries, Berlin, Germany

[2]Department of Geography and Planning, Appalachian State University, Boone, NC, USA

[3]Center for Limnology, University of Wisconsin–Madison, Madison, WI, USA

[4] Department of Natural Resources, Cornell University, Ithaca, NY, USA

*Correspondence to*: Benjamin M. Kraemer (ben.m.kraemer@gmail.com)



**Abstract.** Lakes provide many important benefits to society including drinking water, flood attenuation, nutrition, and recreation. Anthropogenic environmental changes may affect these benefits by altering lake water levels. However, background climate oscillations such as the El Nino Southern Oscillation, and the North Atlantic Oscillation can obscure

long-term trends in water levels, creating uncertainty over the strength and ubiquity of anthropogenic effects on lakes. Here we account for the effects of background climate variation and test for long-term (1992-2019) trends in water levels in 117 globally-distributed large lakes using satellite altimetry data. On average, 27% of water level variation in individual lakes was associated with background climate variation. The relative influence of specific axes of background climate variation on water levels varied substantially across and within regions. After removing the effects of background climate variation on

water levels, long-term water level trend estimates were lower (+1.0 cm year$^{-1}$) than calculated from raw water level data (+1.4 cm year$^{-1}$). However, the trends became more statistically significant in 76% of lakes after removing the effects of background climate variation (the median p-value of trends changed from 0.12 to 0.02). Thus, robust tests for long-term trends in lake water levels which may or may not be anthropogenic will require prior isolation and removal of the effects of background climate variation.  Our findings suggest that background climate variation often masks long-term trends in

environmental variables, but can be accounted for through more comprehensive statistical analyses.

## 1 Introduction:

The water level of a lake is an integrative indicator of local and regional hydrology. By extension, variation in lake water levels through time captures the dynamic nature of the water cycle, particularly when coherent patterns are observed

across many lakes (Molinos et al., 2015). Water level variation is often associated with oscillatory dynamics in Earth's hydroclimate such as the El Niño Southern Oscillation (ENSO) (Ghanbari and Bravo, 2008; Stager et al., 2007), the Pacific Decadal Oscillation (PDO) (Benson et al., 2003; Wang et al., 2010), North Atlantic Oscillation (NAO) (Benson et al., 1998), and the Indian Ocean Dipole (IOD) (Marchant et al., 2007). For instance, water levels in multiple lakes in Poland can increase by 20 cm or more during the positive phase of the NAO (Wrzesiński et al., 2018). When multiple axes of

background climate variation overlap, the effects can be even more intense. For instance, due to strong oceanic temperature anomalies in the late 1990's associated with both the ENSO and the IOD, water levels in eight East African Great Lakes went up by more than 1 meter in less than a year (Mercier et al., 2002). This constituted a combined increase in water storage of more than 266 km$^3$—more than half the volume of Lake Erie.

Human activities can also directly affect lake water levels, creating variation that is independent of background

climate dynamics (Aladin et al., 2009; Pekel et al., 2016; Rodell et al., 2018; Tao et al., 2015). For instance, in the 1970's, the two main inflowing rivers to the Aral Sea were diverted in an attempt to irrigate cotton plantations in Central Asian deserts (Aladin et al., 2009; Micklin, 1988). As a result, the water level in the Aral Sea dropped by 2 m in the first decade following the onset of irrigation, and continued to decline as water use in the watershed intensified. In this case, attribution of water level variation to human activity was robust because an abrupt change in surface water management coincided with





a comparably abrupt change in the water level of downstream lakes. However, attributing water level variation to human activity is more difficult when anthropogenic effects are subtle relative to the effects of background climate variation (Corti et al., 1999). This challenge is especially salient when attributing water level variation to anthropogenic climate change (Hassanzadeh et al., 2012; Rodell et al., 2018), especially at the global scale (Rodell et al., 2018). Indeed, there is an ongoing

debate about the global extent and strength of climate change effects on water level variation in lakes (Muller, 2018).

Climate change can affect water levels through a complex web of forces linking surface temperature with hydrology (Block and Strzepek, 2012; Ramanathan et al., 2001; Rodell et al., 2018). While warming has been observed across the Earth's surface, hydrological responses to warming are highly variable with some areas becoming wetter and others becoming dryer (Greve et al., 2014; Rodell et al., 2018; Wang et al., 2012). However, more than three-quarters of Earth's

land mass has seen no substantial change in total wetness or dryness in response to recent climate change (Greve et al., 2014; Greve and Seneviratne, 2015). Thus, the effects of climate change on water levels may be subtle relative to background climate variation (Jöhnk et al., 2004), calling for statistical approaches that can effectively account for the effects of background climate variation. Analyzing water levels from lakes worldwide may be especially helpful in reducing uncertainty about the potential contribution of background climatic variation to long-term trends.

Here, we build off previous studies focused on specific lake regions (Clites et al., 2014; Molinos and Donohue, 2014; Molinos et al., 2015; Pasquini et al., 2008) and attempt to disentangle the effects of background climate variation from other drivers of water levels in 117 globally distributed lakes using time series of remotely-sensed water levels from 1992 to 2019. We investigate two key areas of uncertainty: (1) whether apparent anthropogenic water level trends in specific lakes can be explained by background climate variation, and (2) whether trends in water levels can be detected in specific lakes

only after accounting for and removing the influence of background climate variation. We use boosted regression trees (BRTs) as a means of removing the effects of background climate variation on water levels in each lake, enabling us to achieve more robust quantification of the  multi-decadal trends which may or may not be anthropogenic. This approach differs from other recently published approaches (Chanut et al., 1988; Molinos et al., 2015) in that it allows for nonlinear relationships, high levels of interactions between axes of climate variation, nonstationarity, and missing data. Finally, we

assess patterns across lakes to generalize about which regions are most strongly influenced by background climate variation. Our overall goal is to better understand the impact of large-scale background climate forcing on lakes in ways that will help communities manage the benefits derived from lakes in the face of global climate dynamics and anthropogenic influences.

## 2 Methods

### 2.1 Overview

The 117 lakes included in our study contain much of the Earth's liquid surface freshwater and a large proportion of freshwater biodiversity (Vadeboncoeur et al., 2011). They span a wide range of lake characteristics including surface area (79.5 to 378,119.3 $km^2$), catchment area (200 to 3,174,000 $km^2$), perimeter (49.4 to 11,478.9 km), latitude (-33.16 to 66.42 °N), and elevation (-74 to 5,216 m above sea level) (Supplementary Material). All continents were represented except for



Antarctica and Australia. According to the HydroLakes database (Messager et al., 2016), at least 24 of these lakes are reservoirs, and 14 of them are natural lakes with some water level regulatory structure such as a dam. The global scope of this analysis builds off similar analyses which aimed to disentangle background climate variation's influence on lake water levels using smaller numbers of lakes from specific regions (Mercier et al., 2002; Molinos et al., 2015) or specific lakes

(Cohn and Robinson, 1975; Stager et al., 2005; Tomasion and Valle, 2000).

Our first objective was to estimate trends in water levels in this global sample of lakes based on average annual water level. We calculated trends using Thiel-Sen non-parametric regression using the 'zyp' package in R. Thiel-Sen slopes represent the median of slopes derived from all pairwise combinations of points in a time series. The statistical significance of each trend (p-value) was calculated using a bootstrapped one sample Wilcox signed-rank test with 1,000 repetitions where

the input data for the test was the complete list of all slopes derived from all pairwise combinations of points in the time series. The number of pairwise slopes used in each repetition of the Wilcox signed-rank test was equal to the number of years of water level data for each lake.

Our second objective was to characterize and account for the effects of background global variation on water level variation. We did this by using boosted regression trees (BRTs) to model water level variation in each lake as a function of

the year, the month of the year, and a large set of global climate indices. We calculated the relative importance of each global climate index in the models for each lake to assess its sensitivity to different axes of background climate variation. We use the partial dependence of water level variation on the year term in the model to reflect the long-term variation in water levels that is not attributable to background climate variation. We repeat the Thiel-Sen slope calculation and p-value calculation based on the partial dependence data for the year term as an estimate of the long-term trend that is not attributable to

background climate variation. This remaining variation could be attributable to human activity, though we cannot draw causal conclusions or distinguish between the various aspects of human activity which can affect water levels.

We derived the background global climate indices for each lake's BRT using principal components analysis (PCA) applied to global variation in monthly earth surface air temperature data through time. This approach is widely applied in the climate sciences to global grids of temperature, pressure, or rainfall data, and is analogous to Empirical Orthogonal Function

(EOF) analysis (Dommenget and Latif, 2008; Hannachi et al., 2009; Kim and Wu, 1999). Temperature time series at each pixel were included as separate variables in the PCA with each time step as a separate observation of those variables. We linearly detrended each principal component (PC) resulting from the PCA using Thiel-Sen non-parametric regression, such that the PC values were equivalent to the residuals from the Thiel-Sen regression. PCs were detrended because long-term trends in the PCs could be considered potentially related to anthropogenic climate change (Stenseth et al., 2003). Thus, the

effect of all PCs on water levels in the BRTs were interpreted as the collective effects of surface air temperatures and their associated hydrological effects on lake levels.

Using the PCs as predictors in each lake's BRT was preferred over using the commonly recognized climate indexes (NAO, ENSO, IOD, etc.) directly in the BRTs for several reasons. First, many commonly recognized global climate indices are collinear which makes them problematic for simultaneous use as predictors, whereas the PCs used here are uncorrelated





by definition. Second, using the commonly recognized global climate indices alone would miss a substantial amount of variation in air temperature that is not periodic but may still drive climate and hydrological variation in lakes. Third, many of the commonly recognized global climate indices are defined subjectively, whereas the PCs used here are identified empirically.

Our modelling approach is based on the recognition that much of the variation in water levels is directly or indirectly driven by global patterns in earth's surface air temperature via the effects of global temperatures on hydrological fluxes (Dommenget and Latif, 2008). This relationship is well-supported because earth's surface air temperature is a key control on earth's hydrological cycle through the Clausius Clapeyron relation which, in turn, drives lake water budgets and water levels (Christensen et al., 2004; Fowler et al., 2007; Nijssen et al., 2001; Tierney et al., 2008). Earth's surface air

temperature may also be a key control on the positioning of atmospheric rivers which can also drive lake water budgets and water levels (Gimeno et al., 2014; Lorenzo et al., 2008). This assumption is empirically well-supported by studies showing earth surface air temperature variation described by ENSO, PDO and IOD is strongly associated with water level variation across the globe (Stager et al., 2007; Tierney et al., 2013; Wang et al., 2010).

      Disentangling the direct and indirect effects of earth surface air temperature variation on water level variation could

be done using a reductionist approach, i.e. by constructing lake hydrological budgets with all of the water inputs and outputs and modelling the effect of human activity on each of those fluxes. But, for most lakes in our dataset, we lack the field measurements required to model the forces linking earth surface air temperature to water level variation for the entire 27-year water level time series (Stenseth et al., 2003). So instead, we use BRTs which mimic the complex web of forces linking earth surface air temperature variation to water level variation. We use BRTs for this purpose because the model structure

accommodates high levels of interactions among predictor variables and mimics the interactive and indirect effects we tried to capture. This approach also differs from other recently published approaches (Chanut et al., 1988; Molinos et al., 2015) in that it allows for nonlinear relationships, non-stationarity, and missing data. We fit BRTs separately for each lake because the influence of a particular climate oscillation could differ across lakes due to geographic forcing, orographic forcing or other local factors (Stenseth et al., 2003). We used backward elimination variable selection techniques to identify the PCs

for each specific lake that, when fit against a training dataset, performed the best when predicting water level variation in a test dataset with which the model had not been fit. Then, we used the resulting BRTs to determine the relative importance of different axes of background climate variation separately for each lake.

## 2.2  Data

Water level data were acquired from the NASA/CNES Topex/Poseidon and Jason satellite missions via the Global Reservoir and Lake Monitoring (G-REALM) project version 2.3 (Crétaux and Birkett, 2006) and can be obtained from: http://www.pecad.fas.usda.gov/cropexplorer/global_reservoir. Although these altimeters were developed to map ocean surface height, they have also been used to detect water level changes in lakes (Birkett, 1995). Only a small subset of the world's lakes can be monitored in this way because the space-borne sensors must pass directly over the lake with sufficient



regularity to produce accurate and complete time series. The US Department of Agriculture (USDA) uses these data to monitor water level variation for many inland water bodies globally. The lakes in this study comprise the 117 lakes with the longest and highest resolution time series. Validation of satellite altimeter data over inland water bodies is typically performed by comparing satellite altimeter measurements and in situ measurements. The root mean squared error of

elevation variations derived from the NASA/CNES Topex/Poseidon and Jason-1 satellite missions is typically ~3 cm for large lakes (Birkett, 1995; Crétaux and Birkett, 2006). Thus, it is justifiable to use water level altimetry in place of in situ gauge measurements (Birkett et al., 2011).

Water levels are typically measured every 10 days, but the exact dates on which water levels are measured vary from lake to lake. To make water level data temporally consistent, we linearly-interpolated each lake's time series to the

daily scale using the 'deseasonalize' and 'zoo,' packages in R (R Core Team, 2017). Monthly averages were calculated so that all lakes had time series of the same interval that also matches the temporal resolution of surface air temperature data used in the PCA. Nine of the 117 water level time series had a data gap from late 2002 through the middle of 2008. The missing data were not estimated, but our analyses treated these lakes in the same way as ones with complete data.

Monthly average land and ocean surface air temperature anomaly data were acquired from the Goddard Institute for

Space Studies (GISS) Land Ocean Temperature Index analysis for a 2 x 2 degree grid with 1200 km smoothing (Hansen et al., 2010). These temperature data are derived from meteorological station observations distributed across the globe, and processed according to methods developed at the National Aeronautics and Space Administration (NASA) (Hansen et al., 2010). They are publicly available at https://data.giss.nasa.gov/gistemp/.

2.3  Principal components analysis

We used PCA to distill the spatial complexity of surface air temperatures for inclusion in each lake's BRT. PCA is an ordination-based statistical tool that converts potentially correlated variables into a set of orthogonal vectors that capture the variation across locations. PCA uses orthogonal linear transformation to identify vectors that account for as much of the total variation in a set of variables as possible. The first PC ($PC_1$) explains the largest percentage of the variation in the

underlying set of variables followed by the second ($PC_2$), third ($PC_3$), and so on. Each succeeding PC is linearly uncorrelated to the others and accounts for as much of the remaining variation as possible. PCA can, therefore, be used to summarize the consistent aspects of time dynamics across space and reduce redundant spatial variation (stemming from spatial autocorrelation and teleconnections) in temperature.

To identify which known oscillations in surface air temperature are related to individual PCs, we calculated a

complete correlation matrix between each PC and all of the 37 major climate indices recognized by NOAA's Earth System Research Laboratory. Monthly time series of major climate indexes were sourced from https://www.esrl.noaa.gov/psd/data/climateindices/list/.  In cases where a PC is highly correlated to one of the major climate indices (PCs), we renamed it with a subscript (e.g., $PC_{ENSO}$) to facilitate interpretation.





2.4 Boosted regression trees (BRTs) and model selection

BRTs were used to model mean monthly water levels in the lakes as a function of year, the month of the year, and a large set of PCs. BRTs were fit using the "dismo" and "gbm" packages (Hijmans et al., 2017) in R (R Core Team, 2017). To cross validate the BRTs for each lake, we fit the model using a training dataset and then used the fit BRT to predict monthly water levels using the PC values from a test dataset. For each lake, we fit six starting models using six different training and test dataset combinations. To get these dataset combinations, we first split the lake level time series for each lake into training and test datasets along its time series using 40-60, 50-50, and 60-40 splits. We split the data along the time series into training and test datasets instead of by randomly selecting observations for the training and test datasets because the data are temporally autocorrelated and we wanted to ensure that the training and test datasets were independent. For each of the three splits, the starting model was fit twice, once using the first part of the split as a training dataset and the second part as a test dataset, and once using the second part of the split as a training dataset and the first part as a test dataset, resulting in a total of six train-test dataset combinations. For each lake's six train-test dataset combinations, we repeatedly refit the starting model after dropping the PC with the lowest relative importance averaged across all six train-test dataset combinations (see explanation of relative importance below) until the starting model had only 2 predictors—the minimum number of predictors allowed in a BRT. Each time a variable was dropped from one of the 6 starting models, we calculated the average change in predicted residual error sum of squares (PRESS; the sum of squares of the prediction residuals calculated using the test data) which resulted from dropping it. Variables which, when dropped from the model, resulted in an average increase in PRESS across the six starting models were selected in that lake's "best BRT." We combined information across models so that the selection of a variable did not depend on the arbitrary decisions of where to split the time series and whether to use the first or second part as the training dataset. Thus, the best BRT included the year, the month, and the combination of PCs from $PC_1$ to $PC_{100}$ which consistently improved its performance in cross validation. The first 100 PCs included all PCs which explained at least 0.1% of the variability in earth surface air temperatures. The first 100 PCs were used in the BRTs in recognition that NOAA distinguishes 37 similar axes of variation and many more potentially go undescribed. This best BRT combination of PCs, specific to each lake, was used for determining the relative importance of the variables selected in the best model. We refit the best BRT for each lake to the full time series to calculate the final relative importance values of each PC. The relative importance of each predictor variable in the model is a function of the frequency with which it was included in the BRT's individual regression trees and the improvement to the model that resulted from its inclusion (Elith et al., 2008). The relative importance of variables that were not selected in each lake's best BRT was considered to be zero.

**3 Results**

We observed considerable variation in water levels within the 117 lakes in our analyses (Fig 1). Prior to accounting for the effects of background climate variation, water levels increased at a median rate of 1.4 cm year⁻¹ (interquartile range: -0.5 cm year⁻¹ to 3.1 cm year⁻¹) (Fig 1). Water levels were decreasing in 35 lakes (29.9%), of which 14 were statistically significant ($p<0.05$ level), and increasing in 82 lakes (70.1%), of which 30 were statistically significant ($p<0.05$ level). In





total, 44 lakes (37.6%) in our analyses had significant trends in water level (Fig 2). Given a significance level of α=0.05, we would expect only 6 of our 117 lakes to show significant trends by chance. However, we observed significant long-term trends in water levels in 44 lakes, far more than predicted based on chance alone (Fig. 2). A comparable disparity was observed across a range of different arbitrary thresholds for statistical significance (i.e., 0.01, 0.05, and 0.1).

5        Changes in water levels from 1992 to 2019 displayed a moderate level of regional consistency in the direction and magnitude of water level trends (Fig 3). In particular, lakes in the Middle East and the Southwest United States tended to have decreasing water levels (Fig 3).  Lakes in Canada and Europe tended to have weak or increasing water level trends. Water level trends in East Asia and Africa were highly variable from lake to lake (Fig 3).

        BRTs performed well for most lakes in our cross-validation and model optimization procedure; the median PRESS
of the best model was 5.0 cm (interquartile range: 2.6-17.2 cm) across lakes. However, our BRTs failed for two lakes with known anthropogenic water level dynamics, both of which are reservoirs on the Mekong River in China where water levels increased >50 m as a result of dam construction in the middle of the time series (Nuozhadu and Xiaowan Dams). Across lakes, the best model included a median of 5 of the 100 PCs (interquartile range: 2-10 PCs) that were fed into our model selection procedure. Remarkably, only 4 out of 100 PCs were never selected in any lake, and the overall frequency of
inclusion across lakes decreased with PC order (Kendall's tau = -0.54, p < 0 .01). Together, the PCs selected for each lake explained a median of 27% of the variance in water levels (interquartile range: 4-44%).

        The relative importance of each specific predictor variable varied substantially from lake to lake. On average, the year and the month were most influential across lakes. On a scale from 0-100 with higher values connoting higher relative importance, year and month had mean relative importance values of 29.1 and 11.1, respectively. $PC_4$, $PC_2$, and $PC_5$ were the
predictor variables with the third (4.3), fourth (4.2), and fifth (3.2) highest mean relative importance in the best performing BRTs (Fig 4). $PC_4$ was moderately correlated with the North Pacific Oscillation (NPO; Kendall's tau = 0.30, p <0.01); $PC_2$ was highly correlated with the Multivariate El Nino Index (MEI; Kendall's tau = 0.72, p <0.01), and $PC_5$ was moderately correlated to the North Atlantic Oscillation (NAO; Kendall's tau = 0.37, p <0.01), so we renamed them here as $PC_{NPO}$, $PC_{ENSO}$, and $PC_{NAO}$ (Fig 5). These three PCs together encompass 18.2% of the variation in surface air temperature anomalies
($PC_{NPO}$ = 5.5%, $PC_{ENSO}$ = 7.8%, and $PC_{NAO}$ = 4.8%). $PC_{NPO}$, $PC_{ENSO}$, and $PC_{NAO}$ were selected in the best models for 42, 43, and 32 lakes, but the direction of their effects differed among waterbodies (Fig 5). Many of the remaining PCs of high mean relative importance across waterbodies (mean relative influence > 1) were only moderately correlated to indices from NOAA (Kendall's correlation coefficient range 0.22-0.34). For instance, $PC_{15}$ was not substantially correlated with any climate index recognized by NOAA (maximum Kendall's tau = 0.17, p <0.01), yet exhibited the 9th highest mean relative
importance in explaining water levels across lakes (Fig 6).

        $PC_{NPO}$, $PC_{ENSO}$, and $PC_{NAO}$ were strongly related to water level variation in lakes around the world but the strength and directionality of those effects were regionally concentrated (Fig 7-9). $PC_{ENSO}$ was positively associated with water levels in Central Canada and Northeast United States and negatively associated with water levels in Sub-Saharan Africa, Northern Europe, South Central Canada, the Equatorial Americas and Central Asia (Fig 7). The strongest effects of $PC_{NAO}$ were


concentrated in North America and East Asia where it had uniformly positive effects on water levels (Fig 8). $PC_{NPO}$ was also selected in the best models of lakes that were regionally concentrated; it was positively associated with water levels in East Africa, Northern Europe, and South East Canada while it was negatively associated with water levels in Northern Canada and Northern East Africa (Fig 9). Not all effects of the PCs on water levels were monotonic—$PC_{NPO}$ had a U-shaped

relationship with water levels in Southeast North America such that both very low and very high values of $PC_{NPO}$ were associated with relatively high water levels (Fig 9).

   After removing the effects of background climate variation on water levels using the fitted BRTs, water level trend estimates were shallower compared to estimates from the original time series (Fig 1). The median water level trend across lakes dropped from +1.4 to +1.0 cm year$^{-1}$ after correcting for background climate variation (Fig 1). Even though they were

lower on average, the trends became more statistically significant in 76% of lakes (Fig 2). Indeed, the median p-value of water level trends across lakes changed from 0.12 to 0.02 after removing the effects of background climate variation (Fig 2). For instance, prior to removing the effects of background climate variation, Lake Mweru in central Africa had an increasing trend (+2.62 cm year$^{-1}$) with a relatively high p-value (0.301). However, after accounting for the effects of background climate variation, the trend was not substantially affected (2.52 cm year$^{-1}$) but was became statistically significant (p-value:

0.009). Based on inspection of the time series of water levels in Lake Mweru, we suspect that the strong peak in lake levels in the late 1990's due to climate oscillations masked the long-term trend (Fig 10). In contrast, prior to removing the effects of background climate variation, Kainji Lake in Nigeria had an increasing trend (+4.32 cm year$^{-1}$) with a relatively low p-value (0.052). However, after accounting for the effects of background climate variation, the trend was moderately lower (+3.52 cm year$^{-1}$) but less statistically significant (p-value: 0.135). Based on inspection of the time series, we suspect that

synergies among background climate variables created a specious appearance of a long-term trend in Kainji Lake (Fig 10).

## 4 Discussion:

   We detected long-term trends in water levels before and after accounting for background climate variation in most lakes in our analyses. The evidence of trends in the majority of lakes belies reports that most of the Earth has experienced no

consistent changes in annual wetness and dryness (Greve et al., 2014; Greve and Seneviratne, 2015). This contrast highlights the potential for waterbody surface levels to serve as integrative metrics of regional water budgets, thereby enhancing our ability to detect hydrological changes.

   Background climate variation had significant effects on water levels in most large lakes between 1992 and 2019. Infrequently, the effects of multiple axes of background climate variation gave rise to the appearance of long-term trends

which became less significant once background climate variation was factored out. But more often, background climate variation masked underlying trends in water levels which were detected when the effects of background climate were factored out (Table S2). Thus, attempting to detect anthropogenic effects on water levels using water level time series without accounting for background climate variation may over- or under- estimate the multi-decadal water level trends in lakes.





The trends in water levels estimated here differed widely among lakes, presumably reflecting the heterogeneity of underlying changes in regional hydrological fluxes. Rising water levels in the majority of lakes may be attributable to increases in precipitation within their watersheds (Bintanja and Selten, 2014; Chadwick et al., 2013; IPCC, 2014; O'Gorman et al., 2012). However, even in watersheds which have experienced increased precipitation, greater inputs of water may be

offset or even exceeded by increases in evapotranspiration (Dorigo et al., 2012; Vinukollu et al., 2011; Vörösmarty et al., 2000; Vörösmarty and Sahagian, 2006) that yield net decreases in water levels.

Not all of the lakes with significant trends in water levels followed the "wet gets wetter and dry gets dryer" pattern that is often predicted to occur with climate change (Wang et al., 2012). According to such predictions, surface water storage would be expected to decrease in many dry mid-latitude and subtropical regions, and to increase at high latitudes and in

humid mid-latitude regions (IPCC, 2014). Several lakes with the strongest decreases in water levels (Aral, Mosul, Powell, Rakshastal, Salton, Tharthar, and Urmia) are indeed located in relatively dry regions (average long-term discharge / watershed area < 5000 $cm^3 s^{-1} km^{-2}$). Furthermore, several lakes with the strongest increases in water levels (Zeyaskoye and Atitlan) are located in relatively wet regions (average long-term discharge / watershed area > 5000 $cm^3 s^{-1} km^{-2}$). However, the intensification of wet-dry contrasts was violated in many places as some lakes in wet regions got dryer (Vermelha,

Winnebago, and Woods) and some lakes in dry regions got wetter (Balkash, Cabora Bassa, Kapachagayskoye, Kariba, and Ulungar). This finding adds to others showing that a range of hydrological fluxes contradict the "wet gets wetter and dry gets dryer" pattern (Byrne et al., 2015). As a specific example from our study, Lake Turkana had a significant increase in water level from 1992 to 2019 despite being in a very dry region. Thus, intensification of contrasts in precipitation may be a useful heuristic for predicting water level trends in some regions, but is clearly inapplicable at the global scale (Greve et al., 2014;

Greve and Seneviratne, 2015).

The changes in water levels in response to the three most important PCs ($PC_{NPO}$, $PC_{ENSO}$, and $PC_{NAO}$) often matched the direction of change predicted from the hydrological changes associated with these particular climate oscillations. For instance, we observed strong negative relationships between $PC_{ENSO}$ and water levels for waterbodies in Sub-Saharan Africa, Equatorial Americas, and South Central Canada, as would be predicted based on studies of the global effects of ENSO on

precipitation (Dai and Wigley, 2000; Ropelewski and Halpert, 1987). We also observed water level changes in North America as a function of $PC_{NAO}$ that are consistent with predictions from observed regional changes in rainfall associated with NAO (Dai et al., 1997). However, we also detected relationships between water levels and PCs that were not consistent with the known hydrological effects of NPO, ENSO, and NAO. For example, we found that $PC_{NPO}$ had a strong effect on water levels in Northern Europe, where NPO is not typically considered to be a major driver of hydrological fluxes.

However, the correlation between $PC_{NPO}$ and the NPO was weak, so the apparent influence of $PC_{NPO}$ could be driven by climate variation that was captured by $PC_{NPO}$ and not the NPO.

Twenty seven percent of the explained variation in water levels (median across lakes) could be attributed to large-scale climate drivers as represented by the first 100 PCs derived from air temperature records. We included higher order PCs because they might have accounted for additional variation in water levels. But we note that higher order PCs encompassed



far less of the variation in surface air temperatures and they were included in far fewer of the lakes' best water level models, so their implications for water levels are far weaker. For example, $PC_{100}$ explained only 0.1% of variation in global surface air temperatures, and was included in our best performing model for 2 lakes albeit with very low explanatory contribution in those cases. Thus, excluding high-order PCs is unlikely to lead to substantial improvements or changes to our conclusions.

The performance of the BRTs might be improved by including lag effects (Hansen et al., 1998; Hidalgo and Dracup, 2003). However, the computation time required to include more PCs or lag effects was prohibitive.

    Our interpretations of the statistical patterns reported herein require several caveats. First, there is substantial debate over which aspect of human activity (e.g. climate change, land use change, dam construction/management) is most important for driving water levels (Gyau-Boakye, 2001; Lenters, 2001; Mercier et al., 2002; Ricko et al., 2011; Wurtsbaugh et al.,

2017). Our modelling approach does not discern whether the trends we calculated are anthropogenic or which aspects of human activity are driving water level trends. Hence, future work is needed to disentangle the various anthropogenic forces which may influence water levels. Detecting anthropogenic water level trends and distinguishing between the effects of different aspects of human activity on water levels could be achieved by including water level dynamics in Earth system models. To date, lake ecosystems are generally oversimplified in such models in which lakes are often assumed to be

relatively static, inert bodies on the landscape.

    Second, we interpreted the trend in the partial dependence values for the year term in each lake's BRT as being potentially anthropogenic. However, we cannot exclude the possibility that anthropogenic water level variation was attributed to background climate PCs themselves in cases where PCs are highly correlated with human activity. Human water extraction may consistently be a larger proportion of total water fluxes during certain phases of the ENSO cycle, for

example. In our BRTs, this aspect of anthropogenic water level variation would have been captured by $PC_{ENSO}$ rather than the year term, and would have been erroneously attributed to background climate variation. However, if the correlation between human water extraction and ENSO changed over time, it would be correctly attributed to the year variable in the BRTs, as we expected.

    Third, we distinguish between anthropogenic climate change and background climate variation in our

interpretations because background climate indices like those used here are generally considered to be modes of natural variation. However, human activity may influence background climate variation as well, perhaps making certain phases of various climate oscillations more likely (Cai et al., 2015; Capotondi and Sardeshmukh, 2017; Timmermann et al., 1999). We partially accounted for this by removing any linear trend through time for each PC prior to fitting the BRTs. Detrending the PCs in this way helped to separate our background climate indices from any ongoing climate change. However, more

complex interactions between climate change and background climate variation would not be removed by this approach, as in a scenario where climate change enhances both positive and negative phases of background climate oscillations yet has no impact on trends in the mean. Research on such complex relationships is still inconclusive (Allen and Ingram, 2002; Cane, 2005; Collins, 2000; Guilyardi et al., 2009; van Oldenborgh et al., 2005), so we performed only linear detrending of PCs. We also recognize that our 27-year time series could also reflect longer-period climate oscillations such as the Multi-Decadal





Atlantic Oscillation, but the limited duration of altimeter-based satellite monitoring of lake levels precludes testing for such influences.

The lakes represented in our study comprise a substantial portion of the global liquid surface freshwater on the planet. Our study includes the ten most voluminous freshwater lakes on Earth's surface (Baikal, Tanganyika, Superior, Michigan,

Huron, Malawi, Victoria, Great Bear, Ontario, Great Slave), which collectively contain more freshwater (total 80,241 km$^3$) than has been withdrawn from the environment by humans at the global scale over the last 20 years (Food and Agriculture Organization of the United Nations, 2016). But the lakes in this study are not representative of all lakes, which tend to have smaller surface area and shallower maximum depths on average. Thus, the relative importance of background climate oscillations in the remainder of lakes other than the 117 large lakes studied here over the last 27 years remains uncertain and

should be investigated further.

Our modelling approach could be widely applied to disentangle the effects of background climate on other hydrological changes including streamflow and pan evaporation rates. The novel statistical method presented here using BRTs could be used to describe or factor out the effects of long-term variation in background climate variation on a wide variety of environmental variables including fires, floods, heatwaves, and droughts—all of which have been shown to be

sensitive to climate teleconnections (Chen et al., 2016; Lau and Kim, 2012; Stenseth et al., 2003). Instead of using common climate indices, we encourage the use of the complete set of PCs calculated here, because important climate oscillations would otherwise be missed. To illustrate this point, the PC with the 9th highest mean relative importance in explaining water level variation was not strongly correlated with any of the reported NOAA climate indices (Fig 7).

Wherever water levels are affected by background climate and human activity, there is the potential to affect lake

ecosystems and the benefits that humans derive from them (Clites et al., 2014). More than two billion people live in water stressed regions of the world where human demand for surface freshwater exceeds the available supply (Mekonnen and Hoekstra, 2016; Vörösmarty et al., 2000, 2010). Meeting the competing demands for surface freshwater, especially in water scarce regions and in the face of anthropogenic environmental change, is a key challenge for society. Our capacity to disentangle the effects of background climate oscillations on water levels is key to sustaining our freshwater resources,

especially in the face of climate change (Clites et al., 2014; Gronewold et al., 2013). By applying this BRT statistical approach, we partially disentangled the effects of background global climate indices on water levels. Many of the large lakes in our analyses were remarkably resilient to long-term changes from 1992 to 2019. Thus large lakes may be an increasingly important resource in the face of water scarcity in the future. Abrupt changes in water levels in large lakes remain possible due to human activities and climate change, but our analyses suggest that we have not yet crossed such thresholds in many of

Earth's largest lakes.

## 5 Conclusions:

On average, water levels in the world's large lakes are increasing but are highly variable from lake to lake. Background climate variability often masks these long-term trends in water levels and occasionally gives rise to the





appearance of false trends that wane after background climate variation is factored out. Background climate variation alone can explain a large proportion of water level variability in lakes worldwide due to the strong influence of earth surface air temperatures on lake levels via climate-lake level teleconnections. These findings highlight further opportunities to investigate the specific mechanisms that couple climate and lake levels. The novel statistical method presented here using

BRTs could be used to describe or factor out the effects of long-term variation in background climate variation on a wide variety of environmental variables including fires, floods, heatwaves, and droughts. Tests for long-term trends in environmental variables which may or may not be anthropogenic will likely benefit from prior isolation and removal of the effects of background climate variation using this method.

**Acknowledgements:**

The lead author is grateful for support from the IGB Leibniz Institute for Freshwater Ecology and Inland Fisheries through their international visiting scholars program and from the German Research Foundation within the LimnoScenES project (AD 91/22-1). We are also grateful for financial support from the John D. and Catherine T. MacArthur Foundation grants G-108015-0 and G-1609-151200 to Appalachian State University, and for further support from the Packard Fellowship in

Science and Engineering.

**Data Availability Statement:**

Water level data products are courtesy of the USDA/NASA G-REALM program which can be found at https://www.pecad.fas.usda.gov/cropexplorer/global_reservoir/. All GISS data can be found at

https://data.giss.nasa.gov/gistemp/.

**Code Availability Statement:**

All code used in the production of this manuscript including data analysis and figures are published in the Zenodo online repository with DOI: 10.5281/zenodo.3363187

**Author Contributions:**

BK designed the study, developed the model code, and performed the analyses. BK prepared the initial manuscript. BK, RA, AS, and PM contributed to the manuscript's revision and editing.

**Competing Interests:**

The authors declare that they have no conflict of interest.





**Figures:**

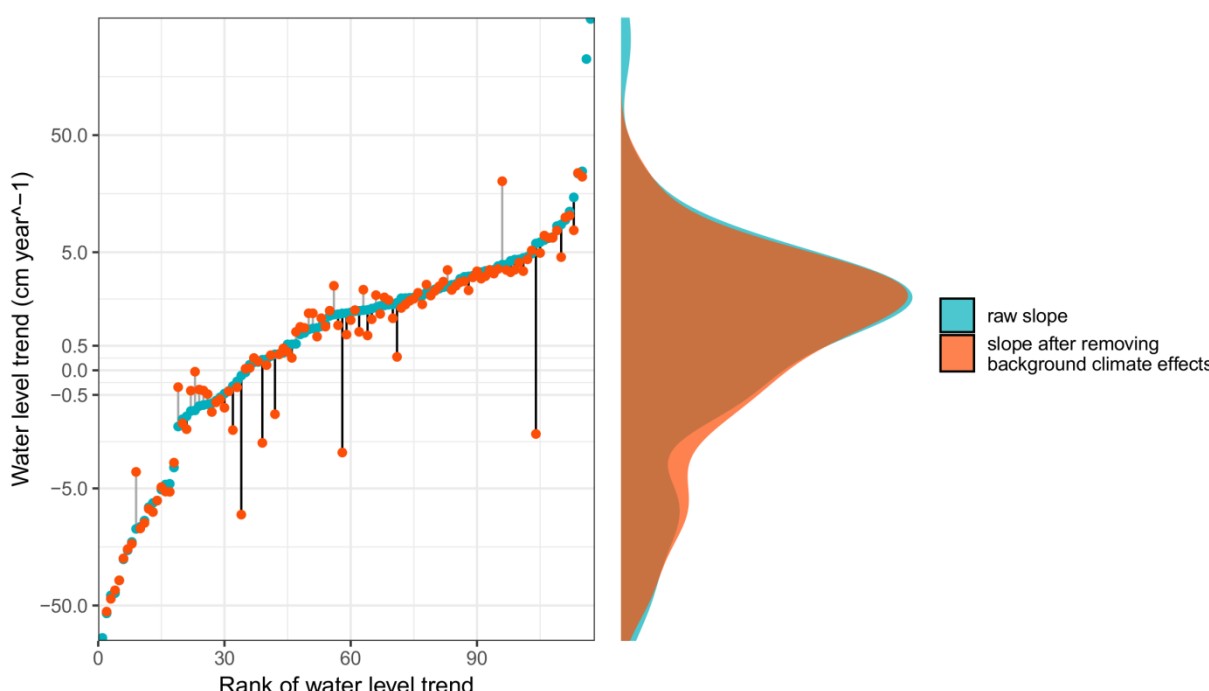

Figure 1: Water level trends in 117 globally distributed large lakes and the change in each lake's water level trend which resulted from removing the effects of background climate variation. After removing the effects of background climate variation on water levels, long-term water level trend estimates (grey) were slightly more conservative overall compared to rates calculated from the raw water level data (black). The median value of water level trends across lakes changed from +1.4 cm year$^{-1}$ to +1.0 cm year$^{-1}$ after removing the effects of background climate variation.





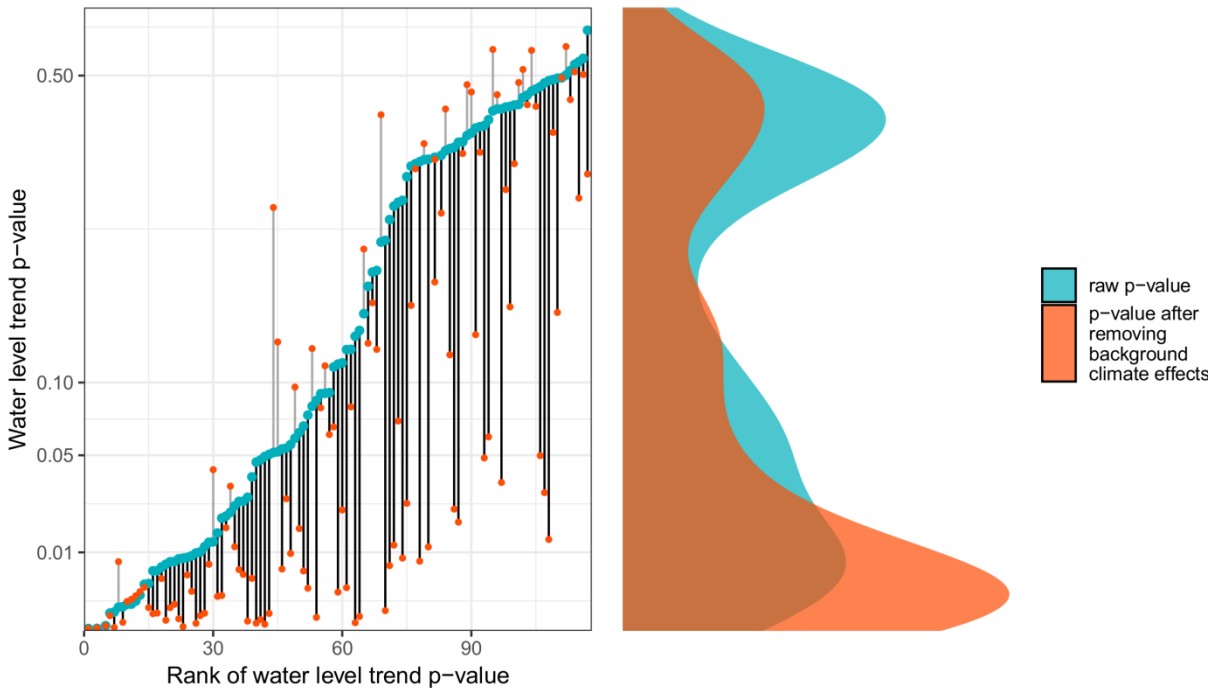

Figure 2: The change in each lake's water level trend p-value which resulted from removing the effects of background climate variation. The trends became more statistically significant in most (76%, blue lines) but not all lakes (24%, red lines) after accounting for the effects of background climate variation. The median p-value of water level trends across lakes changed from 0.12 to 0.02 after removing the effects of background climate variation.



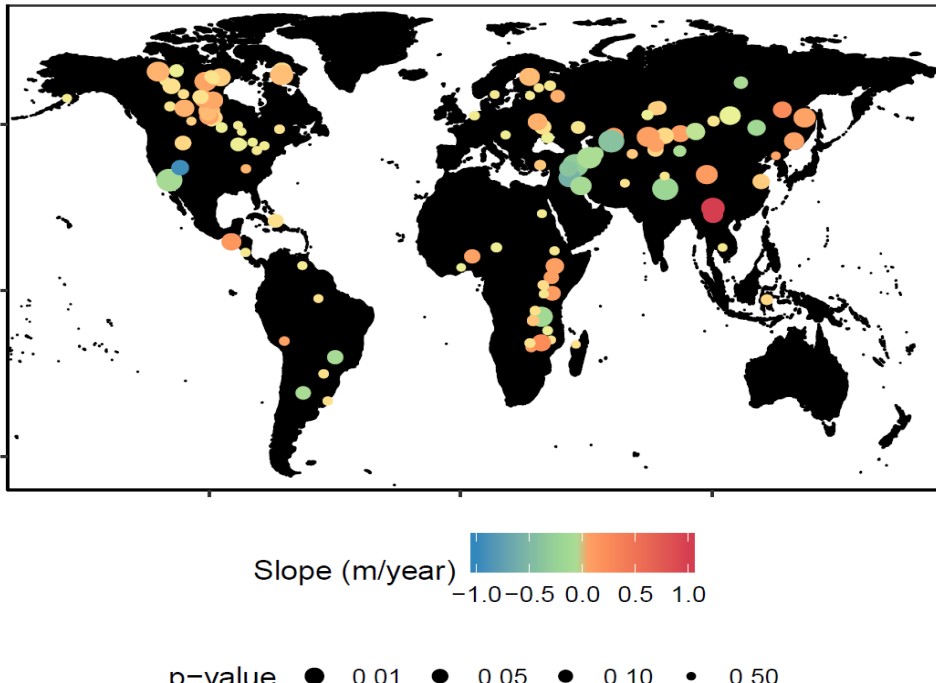

Figure 3: Long-term trends in water levels prior to removing the effects of background climate variation. Some lakes including those in the southwest United States, parts of Africa, and the Middle-East show regionally-consistent patterns in water levels.





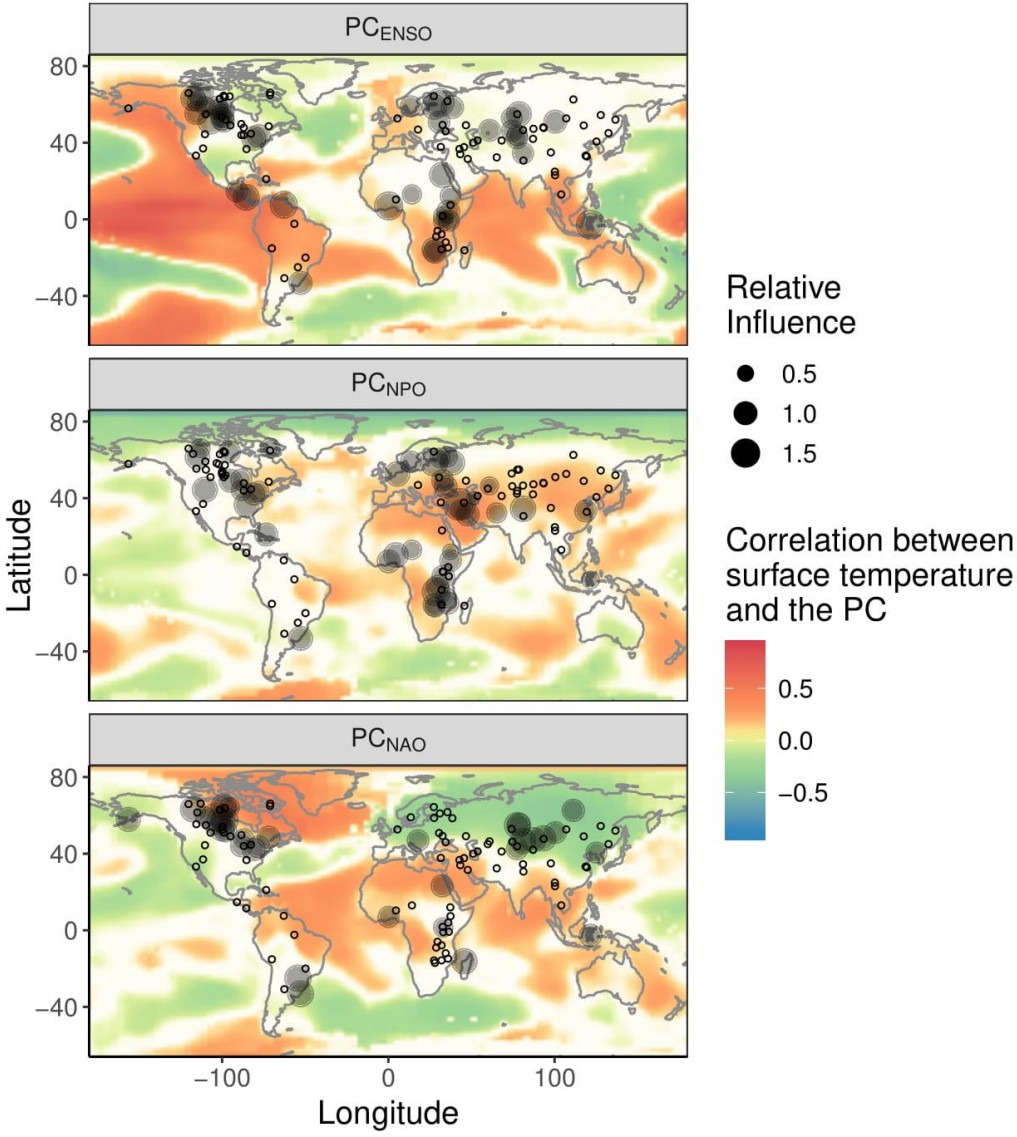

Figure 4: The relative influence of different axes of background climate variation (PCs) on water level variation. Empty dots represent lakes for which the PC was not selected in its "best model." The opacity of each colored pixel in each map is related to the significance of the correlation between the PC and temperature at that pixel with less significant correlations appearing white.





Figure 5: Time series of background climate variation for the three PCs that were most influential in the best models of water level variation on average across lakes. PCs ($PC_{ENSO}$, $PC_{NAO}$, and $PC_{NPO}$) and their corresponding climate indexes (ENSO, NAO, and NPO) are transformed to their z-scores so that they can be more easily compared on the same, unit-less scale. The

5    dots represent raw values and the lines represent locally-weight scatter plot (LOWESS)-smoothed time series.



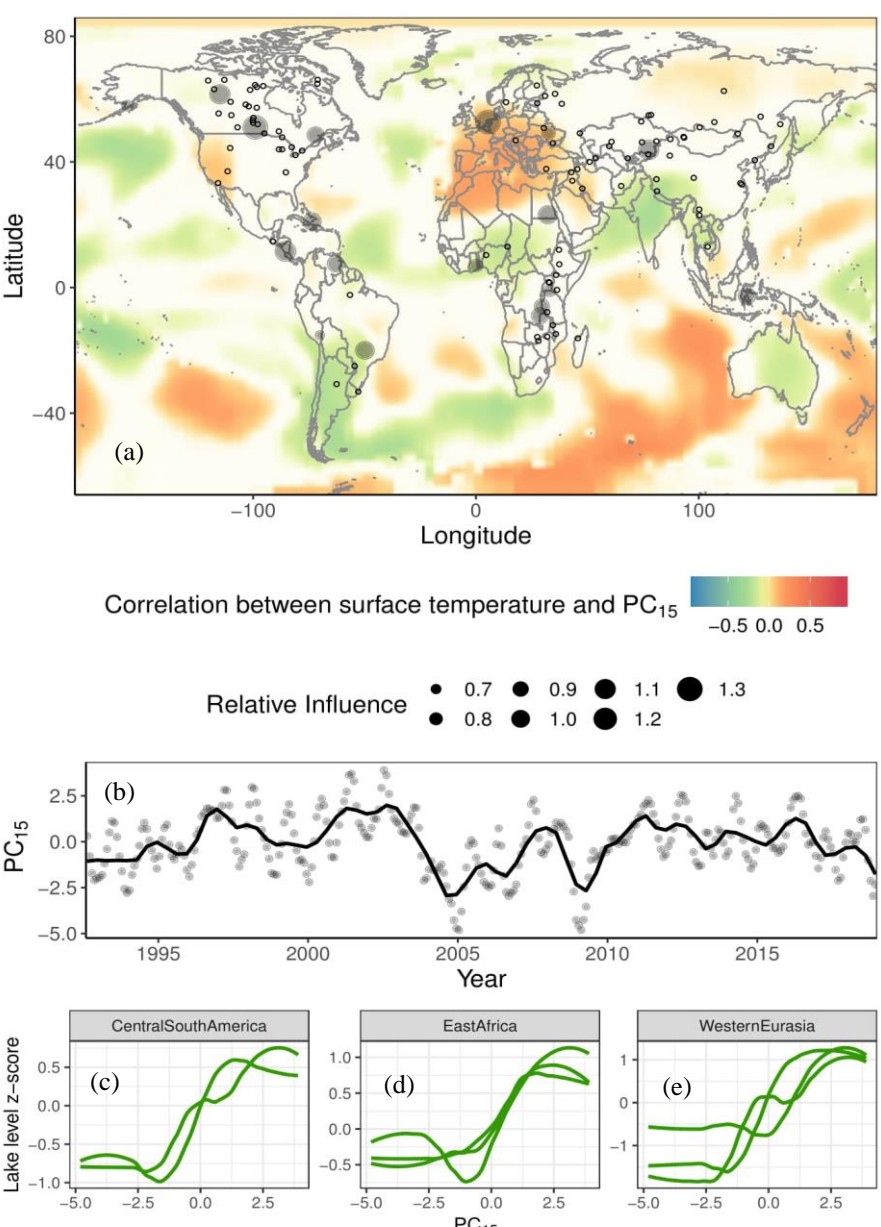

Figure 6: $PC_{15}$ was important for explaining water levels but was not strongly correlated with any of the climate indices recognized by NOAA. In panel (a), Empty dots represent lakes for which the PC was not selected in its "best model." The opacity of each colored pixel in each map is related to the significance of the correlation between the PC and temperature at that pixel with less significant correlations appearing white. In panel (b), the dots represent raw values of $PC_{15}$ and in panels (b-e) the line represents a locally-weight scatter plot (LOWESS)-smoothed values.





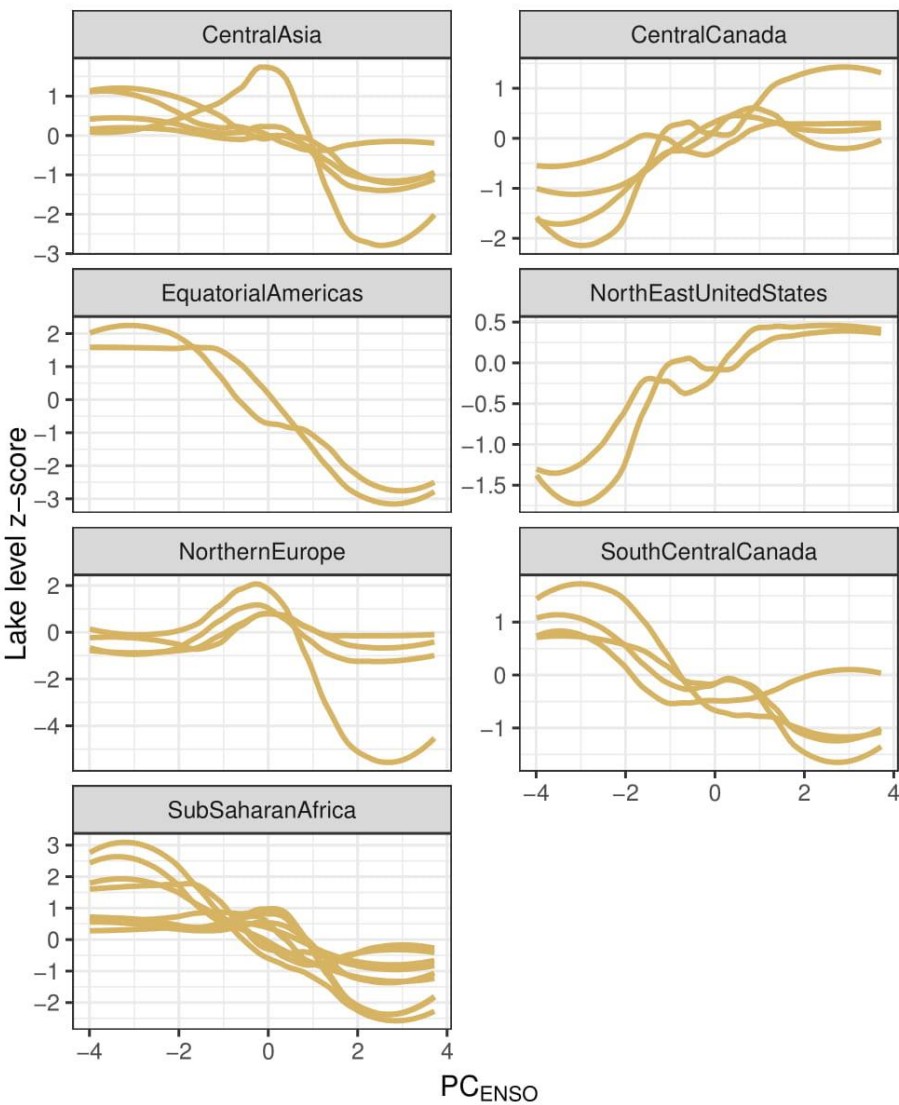

Figure 7: Regional consistently in the directionality of PC$_{ENSO}$ effects on water level variation. Each line represents a LOWESS-smoothed relationship between water level variation in a specific lake and PC$_{ENSO}$. Water levels have been transformed into z-scores so that they may all be plotted on the same axis.





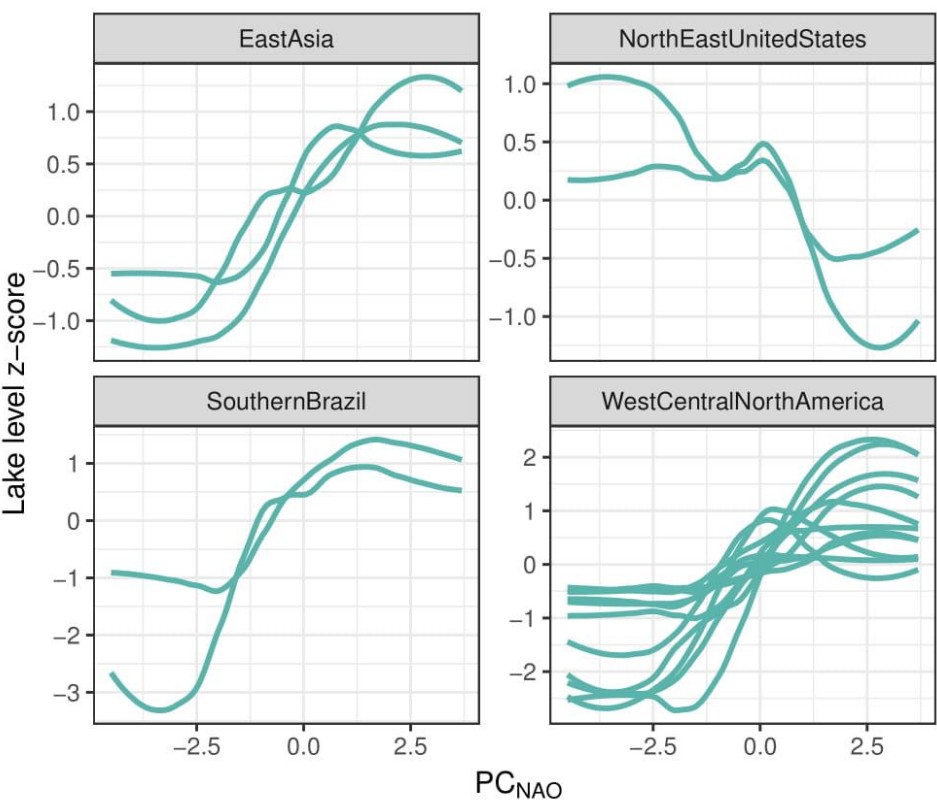

Figure 8: Regional consistently in the directionality of $PC_{NAO}$ effects on water level variation. Each line represents a LOWESS-smoothed relationship between water level variation in a specific lake and $PC_{NAO}$. Water levels have been transformed into z-scores so that they may all be plotted on the same axis.







Figure 9: Regional consistently in the directionality of $PC_{NPO}$ effects on water level variation. Each line represents a LOWESS-smoothed relationship between water level variation in a specific lake and $PC_{NPO}$. Water levels have been transformed into z-scores so that they may all be plotted on the same axis.



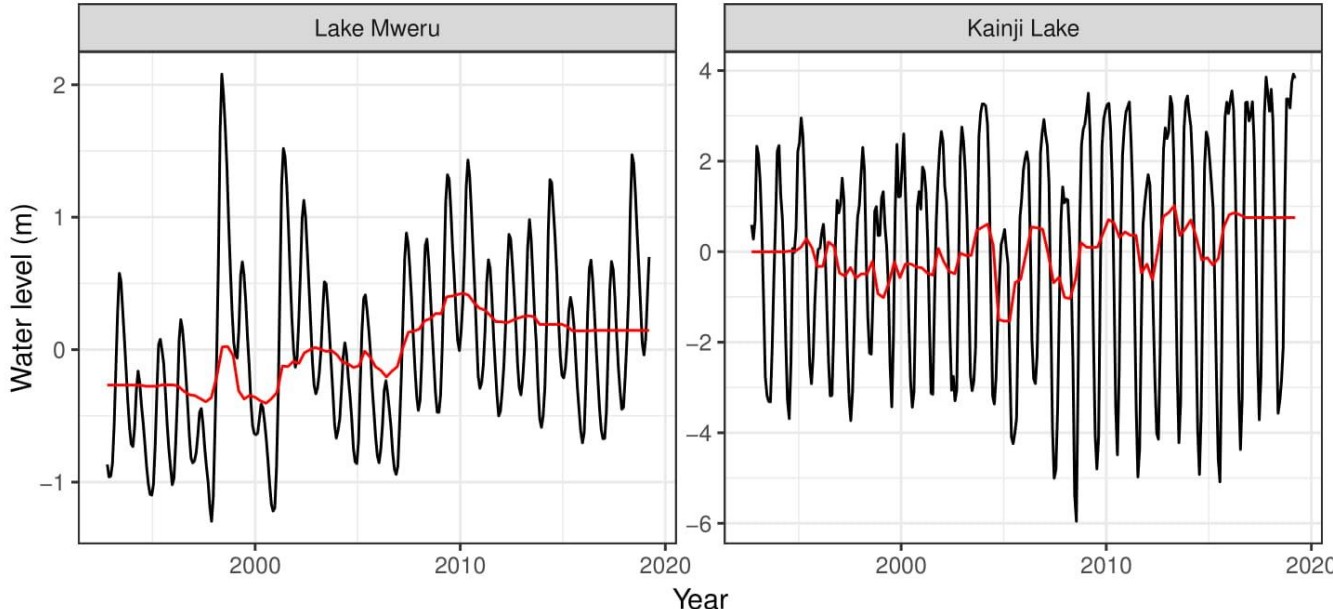

Figure 10: Water level time series for two tropical African lakes which show the potential for (a) background climate variation
to mask long-term trends and (b) the potential for climate variation to give rise to the false appearance of long-term trends. The
water level trend in Lake Mweru on the Zambia-Democratic Republic of Congo border became more significant after
accounting for background climate variation and the water level trend in Kainji Lake in western Nigeria became less
10 significant after accounting background climate variation. Raw data are shown here but all trends were calculated based on
annual averages. The black line is the raw water level data and the red line is the partial dependence of water level on the year.



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
