# Peer review of "Worldwide lake level trends and responses to background climate variation"

_Hydrology and Earth System Sciences, 2019_

## Referee Comment (RC1) · Anonymous Referee #1 · 10 Nov 2019

This study reveals the influence of background climate oscillations such as the El Nino Southern Oscillation, and the North Atlantic Oscillation on worldwide lake level changes (117 large lakes). The authors compared the lake level trends before and after removing of background climate variation. The idea of this study is novel. However, this manuscript still needs improve its quality for publications.

In addition, I am a remote sensing hydrology scientist, but not a hydroclimatology scientist. The most content of this manuscript is about climatic effects on lake level changes. I suggest to add a review from hydroclimatology science for objective evaluation of this manuscript, and for some comments to improve the quality of this study.

My very limited comments are as follows:

1) The method of this study can be displayed by a tree-structure such as Figure 3a in http://dx.doi.org/10.1002/2015GL066235 ?

2) Figures 1 and 2 can be combined together, as Figure 1 shows the lake level characters and Figure 2 shows the corresponding p-value?

3) Figure color: it is usually a warm color (red) for water loss and cold color (blue) for water gain. In addition, a title in head of each figure could be removed? You can use such as "a) PCENSO" for Figure 4a in upper left, and others are similar.

4) L20: "The water level of a lake is an integrative indicator of local and regional hydrology." Two related citations are suggested here: http://dx.doi.org/10.1016/j.rse.2011.03.005; http://dx.doi.org/10.1007/s10712-016-9362-6

5) L25: 1 meter to 1 m.

6) L30: "On average, water levels in the world's large lakes are increasing but are highly variable from lake to lake." This conclusion is not consistent with a recent paper "Recent global decline in endorheic basin water storages" in Nature, https://doi.org/10.1038/s41561-018-0265-7. Could you explain which is right and reason?

---

## Referee Comment (RC2) · Anonymous Referee #2 · 12 Dec 2019

Page 2, Ln 25-27: Please double check this calculation. The total area of the eight East Africa Great Lakes is ∼ 152410 km^2. The water volume increase associated with 1 meter of water level increase is ∼ 152 km^3.

Page 4, Ln 34: "Collinear" is a too strong word here. I would use "correlated".

Page 5, Ln 30-32: Altimeter observed water level changes for global major lakes are also available from the CNES Hydroweb (http://hydroweb.theia-land.fr/). There are notably large differences between the G-REALM and Hydroweb solutions. The G-REALM solutions appear to show larger biases (based on preliminary comparisons in Caspian Sea and Lake Victoria). This is not to ask the authors to redo the analysis using the Hydroweb solutions, but to remind them the potential issues with the G-REALM solutions.

It will be helpful to show a distribution map of the 117 lakes. The current Figure 3 does not really suit the purpose.

Page 6, Ln 6: water level altimetry -> altimetry water level observations

Page 6, Ln 29-33: The authors claimed that they calculated a complete correlation matrix between each PC and all of the 37 major climate indices recognized by NOAA's Earth System Research Laboratory. Some of the indices (e.g., EA/WR and TPI/IPO) from the ESRL (https://www.esrl.noaa.gov/psd/data/climateindices/list/) do not cover the recent time spans. How did the authors deal with those indices?
* * *

---

## Referee Comment (RC3) · Anonymous Referee #3 · 22 Dec 2019

The manuscript analyses the water level variations for 117 globally-distributed lakes and surface air temperature in order to disentangle the effects of climate variation and anthropogenic activities on lake variations. The authors use boosted regression trees (BRTs) to model water level as a function of time and the PCs of temperature. Then those most influential PCs are correlated with climate indices. The topic is highly relevant and will provide important contributions to current climate and anthropogenic impacts on inland water bodies, e.g. lakes.

General Comments

In general, the manuscript is generally well organized and easy to read. But the method section is a bit redundant and needs to be clarified.

[Figure]

The main assumption is that the pattern of global lake changes is somehow driven by global patterns of earth surface air temperature. But the results (27% of water level variation was associated with background climate variation) do not support this assumption very well. In my opinion, a better re-designation of the study objectives is needed.

The authors stressed that a novel statistic method BRT is used, but in my opinion, more details are needed. On top of that, some justifications and clarifications are needed as pointed out below. I hope my comments are useful for authors to improve the manuscript.

Specific comments

P4L6: "... based on average annual water level..." why and how is annual water level used?

P4L13-21: Does this consider the nonstationary of time series induced by climate?

P4L15: Here the climate indices should be renamed or clarified to be distinguishable from those called climate index, e.g. ENSO, IOD, NAO, etc. It is a bit confusing.

P4L22-31: Why are the PCs detrended instead of temperature detrended? How can detrended PCs remove anthropogenic effect?

P4-P5: I think the overview of method should be simplified and move some of the description to specific methodologies.

P6, 2.3: Regarding PCA, a bit more details maybe help to reproduce the results presented in the manuscript. Why 100 PCs are used? Personally, I think those beyond 20 are very weak and probably mainly noises. Is a constant number of PCs used in the BRT for all lakes? I think that not all PCs are statistically significant in the regression. Does BRT take this into consideration?

P6L3: "the longest and highest resolution time series". How long are they? Do they

vary for different lakes? What does it mean when you say "highest resolution"? Be specific.

P6L5: "... is typically ∼3 cm for large lakes...". This is a bit overstated. The best case is around 3 cm. Please double check and revise.

P6L9: "...linearly-interpolated each lake's time series...". Why not use a model considering annual and semi-annual variations given that you have quite long time series? I suggest using the approach used in previous studies, e.g. (Kleinherenbrink et al., 2015; Villadsen et al., 2014)

P7 2.4: The description of statistical method, BRT should be expanded instead of just saying some R packages are used given that you state this method is novel in your study.

P9L26: Lake water level definitely is an integrative metrics of regional water budget. I think this sentence should be rephrased.

P9L26: "significant effects ... in most lakes", is this overstated?

P11L28-29: As pointed above, I doubt that detrending the PCs helps to sperate background climate indices from ongoing climate change. Instead, I would first detrend the time series of temperature, and the perform PCA.

References

Kleinherenbrink, M., Lindenbergh, R. C. and Ditmar, P. G.: Monitoring of lake level changes on the Tibetan Plateau and Tian Shan by retracking Cryosat SARIn waveforms, J. Hydrol., 521, 119–131, doi:10.1016/j.jhydrol.2014.11.063, 2015.

Villadsen, H., Andersen, O. B. and Stenseng, L.: Annual cycle in lakes and rivers from CryoSat-2 altimetry - The Brahmaputra river, Int. Geosci. Remote Sens. Symp., 894–897, doi:10.1109/IGARSS.2014.6946569, 2014.

470, 2019.

---

## Author Comment (AC1) · 31 Jan 2020

**Thank you very much to the reviewer for their thoughtful review and commentary. It has lead to several substantial improvements to the manuscript. We have responded to each of your comments in bold below.**

Reviewer 1:

This study reveals the influence of background climate oscillations such as the El Nino Southern Oscillation, and the North Atlantic Oscillation on worldwide lake level changes (117 large lakes). The authors compared the lake level trends before and after removing of background climate variation. The idea of this study is novel. However, this manuscript still needs improve its quality for publications. In addition, I am a remote sensing hydrology scientist, but not a hydroclimatology scientist. The most content of this manuscript is about climatic effects on lake level changes. I suggest to add a review from hydroclimatology science for objective evaluation of this manuscript, and for some comments to improve the quality of this study.

My very limited comments are as follows:

1) The method of this study can be displayed by a tree-structure such as Figure 3a in http://dx.doi.org/10.1002/2015GL066235 ?

**We welcomed the suggestion of including a regression tree in our analysis. We opted for a boosted regression tree analysis because it tends to outperform simple regression trees according to a variety of performance metrics. This additional analysis will be included in our revision as supplementary material.**

2) Figures 1 and 2 can be combined together, as Figure 1 shows the lake level characters and Figure 2 shows the corresponding p-value?

**We are grateful to the reviewer for this suggestion. We have combined Figures 1 and 2 into the same figure. We agree that the content of Figures 1 and 2 were related enough to justify their merging.**

3) Figure color: it is usually a warm color (red) for water loss and cold color (blue) for water gain. In addition, a title in head of each figure could be removed? You can use such as "a) PCENSO" for Figure 4a in upper left, and others are similar.

**Thanks to the reviewer for these suggestions. We reversed the colour schemes so that red stands for water loss and blue stands for water gain. This improved colour scheme is more intuitive and more in agreement with norms in the literature. We also removed the title heads on each of the figure panels and replaced them with over-plot panel labels as suggested. This change reduces the clutter in the figures.**

4) L20: "The water level of a lake is an integrative indicator of local and regional hydrology." Two related citations are suggested here: http://dx.doi.org/10.1016/j.rse.2011.03.005; http://dx.doi.org/10.1007/s10712-016-

**Thanks for these suggested citations, we enjoyed reading them and have added citations to them in the manuscript.**

5) L25: 1 meter to 1 m.

**Thanks for this suggestion. We have corrected it.**

6) L30: "On average, water levels in the world's large lakes are increasing but
are highly variable from lake to lake." This conclusion is not consistent with a recent paper
"Recent global decline in endorheic basin water storages" in Nature,
https://doi.org/10.1038/s41561-018-0265-7. Could you explain which is right and reason?

**The difference can be reconciled by differences in the methods used. The recent article in Nature Geoscience used shorter time series (only 14 years), analysed terrestrial water storage generally as opposed to lake water storage specifically, and are based on water volumes not water levels. We have added a brief discussion of this topic to the manuscript and cited the recent paper that was mentioned.**

---

## Author Comment (AC2) · 31 Jan 2020

**Thank you very much to the reviewer for their thoughtful review and commentary. It has lead to several substantial improvements to the manuscript. We have responded to each of your comments in bold below.**

Reviewer 2:

Page 2, Ln 25-27: Please double check this calculation. The total area of the eight East Africa Great Lakes is ~ 152410 km^2. The water volume increase associated with 1 meter of water level increase is ~ 152 km^3.

**We have double checked this calculation and found it to be correct. The line states that "water levels in the eight East African Great Lakes went up by more than 1 m." The actual increases are listed below:**

**Lake Tanganyika: Surface area (32000 km2) * Water level increase (1.7 m) = 54.4 km3**
**Lake Victoria: Surface area (69000 km2) * Water level increase (1.4 m) = 96.6 km3**
**Lake Mweru: Surface area (4500 km2) * Water level increase (2.6 m) = 11.7 km3**
**Lake Kyoga: Surface area (1700 km2) * Water level increase (1.9 m) = 3.23 km3**
**Lake Turkana: Surface area (6750 km2) * Water level increase (3.8 m) = 25.65 km3**
**Lake Rukwa: Surface area (3050 km2) * Water level increase (2.4 m) = 7.32 km3**
**Lake Malawi: Surface area (30000 km2) * Water level increase (1.2 m) = 36.0 km3**
**Lake Kariba: Surface area (5200 km2) * Water level increase (5.9 m) = 30.68 km3**

**Total = 265.58 km3**

**This total is in agreement with the total offered in the manuscript.**

Page 4, Ln 34: "Collinear" is a too strong word here. I would use "correlated".

**Thanks for this clarifying suggestion. We have replaced the word "collinear" with "correlated."**

Page 5, Ln 30-32: Altimeter observed water level changes for global major lakes are also available from the CNES Hydroweb (http://hydroweb.theia-land.fr/). There are notably large differences between the G-REALM and Hydroweb solutions. The G-REALM solutions appear to show larger biases (based on preliminary comparisons in Caspian Sea and Lake Victoria). This is not to ask the authors to redo the analysis using the Hydroweb solutions, but to remind them the potential issues with the G-REALM solutions.

**Thanks for this reminder of the potential issues with the G-REALM solutions. Despite the trade-offs as mentioned by the reviewer, we decided to use G-REALM data because it covers more lakes over longer time periods.**

It will be helpful to show a distribution map of the 117 lakes. The current Figure 3 does not really suit the purpose.

**Thanks to the reviewer for this suggestion. We are not sure what a distribution map would show which is not already shown in Figure 3 but would be happy to include one regardless in the supplement to our revision.**

Page 6, Ln 6: water level altimetry -> altimetry water level observations

**Thanks for this suggestion. It has been changed as requested by the reviewer.**

Page 6, Ln 29-33: The authors claimed that they calculated a complete correlation matrix between each PC and all of the 37 major climate indices recognized by NOAA's Earth System Research Laboratory. Some of the indices (e.g., EA/WR and TPI/IPO) from the ESRL (https://www.esrl.noaa.gov/psd/data/climateindices/list/) do not cover the recent time spans. How did the authors deal with those indices?

**For the indices that were not calculated continuously to the present, we calculated correlation coefficients over the longest timespan that was available for each PC and its paired index. This has now been clarified in the methods section with the following text:**

**"For indices that aren't updated to the present, we calculated the correlation over the longest time period over which each major climate index was available."**

---

## Author Comment (AC3) · 31 Jan 2020

**Thank you very much to the reviewer for their thoughtful review and commentary. It has lead to several substantial improvements to the manuscript. We have responded to each of your comments in bold below.**

Reviewer 3:

The manuscript analyses the water level variations for 117 globally-distributed lakes and surface air temperature in order to disentangle the effects of climate variation and anthropogenic activities on lake variations. The authors use boosted regression trees (BRTs) to model water level as a function of time and the PCs of temperature. Then those most influential PCs are correlated with climate indices. The topic is highly relevant and will provide important contributions to current climate and anthropogenic impacts on inland water bodies, e.g. lakes.

**We are grateful for these comments summarizing the manuscript.**

General Comments

In general, the manuscript is generally well organized and easy to read. But the method section is a bit redundant and needs to be clarified. The main assumption is that the pattern of global lake changes is somehow driven by global patterns of earth surface air temperature. But the results (27% of water level variation was associated with background climate variation) do not support this assumption very well. In my opinion, a better re-designation of the study objectives is needed.

**Thanks to the reviewer for this suggestion on how to clarify and improve the methods section. 27% of the variability was, in fact, attributable to the the PCs representing de-seasonalized and de-trended global variation in earth surface temperatures. However, the seasonal component of the model explained an additional 5%, and the long-term trends component explained an additional 13% of the variation. So, in sum, we were able to explain 45% of the variation in water levels on average across lakes. In this context, the importance of the PCs is more apparent because they explained more than 5 times the amount of lake level variation than background seasonality (27% > 5 x 5%). This has now been clarified in the results section and in the abstract with the following text:**

**In the abstract: "On average, 27% of water level variation in individual lakes was associated with background climate variation with an additional 18% explained by seasonal variation and the long-term trend."**

**In the results section: "Together, the PCs selected for each lake explained an average of 27% of the variance in water levels (interquartile range: 4-44%) compared to 5% explained by seasonality and 13% explained by the long-term trend."**

The authors stressed that a novel statistic method BRT is used, but in my opinion, more details are needed. On top of that, some justifications and clarifications are needed as pointed out below. I hope my comments are useful for authors to improve the manuscript.

**Thanks very much for the comments below on how to further clarify the methods section. Our responses are detailed below.**

Specific comments

P4L6: ". . . based on average annual water level. . ." why and how is annual water level used?

**Thanks for pointing out the need to clarify here. We have added the following text to the manuscript: methods section:**

**"Annual water levels were used in the trend analysis instead of raw water levels so that the trend residuals would not be serially autocorrelated."**

P4L13-21: Does this consider the nonstationary of time series induced by climate?

**Yes it does. All effects in the model including the effects of the PCs and the effects of season are allowed to change from year to year. For example, the effect of PCenso on water level variation might be very important in driving water levels in some years for a specific lake but not in others. We have added the following text to the manuscript: methods section:**

**"The main advantages of BRTs over other statistical models is that they have higher predictive performance, do not require data transformation or outlier elimination, automatically handle complex nonlinear relationships and interactions, and allow for many types of predictor variables and partial missing data. Through these interactions, the BRT allows for non-stationarity of the timeseries (e.g. the effect of each PC is allowed to change over time)."**

P4L15: Here the climate indices should be renamed or clarified to be distinguishable from those called climate index, e.g. ENSO, IOD, NAO, etc. It is a bit confusing.

**Thanks for pointing out the need to clarify here. We have added the following text to the methods section:**

**"In cases where a PC is highly correlated to one of the major climate indices (PCs), we renamed it with a subscript (e.g., $PC_{ENSO}$) to facilitate interpretation. In all remaining cases, PCs were named with a numeric subscript which matched their order in the PCA (e.g. "PC1")**

P4L22-31: Why are the PCs detrended instead of temperature detrended? How can detrended PCs remove anthropogenic effect?

**Thanks for this insightful comment. In our revision, we detrended the temperature data prior to the PCA as suggested by the reviewer. Detrending the temperature data also removed the long-term trends from the PCs. We would also add that detrending in this way as suggested by the reviewer predominantly removed the effect of anthropogenic climate change. However, the broader "anthropogenic effect" which includes water extractions and**

**reservoir construction have not been removed from the PCs. We have now clarified this throughout our revision of the methods section and the discussion section.**

P4-P5: I think the overview of method should be simplified and move some of the description to specific methodologies.

**Thanks for this suggestion. In our revision, we have moved much of the text from the "overview" section to the subsequent relevant topic sections.**

P6, 2.3: Regarding PCA, a bit more details maybe help to reproduce the results presented in the manuscript. Why 100 PCs are used? Personally, I think those beyond 20 are very weak and probably mainly noises. Is a constant number of PCs used in the BRT for all lakes? I think that not all PCs are statistically significant in the regression. Does BRT take this into consideration?

**We used an excess number of PCs in the model fitting to ensure that we captured all of the important background climate drivers. But with that said, in our revision, we removed all PCs which were indistinguishable from random noise using a Box-Ljung test (alpha = 0.01). Based on the Box-Ljung test, we eliminated 5 PCs from our BRT model selection procedure. We also include a rigorous evaluation of the importance of each of the PCs in the model through a model selection process. Through the model selection process, if the inclusion of a PC in the BRT did not lead to an improvement in the model's performance during cross validation, it was removed from the model. This is a more rigorous way to exclude variables than exclusion based on "statistical significance."**

P6L3: "the longest and highest resolution time series". How long are they? Do they vary for different lakes? What does it mean when you say "highest resolution"? Be specific.

**We are grateful to the reviewer for bringing up these needed clarifications. In our revision, we have updated the time series for each lake so that it begins in 1992 and ends in 2020. We have included the exact number of observations for each lake in the supplementary Table S3. By "longest" we mean greater than 27 years, and by "highest resolution," we mean, "the greatest number of observations per unit time." This has been clarified in the line referenced by the reviewer. In our revision, the sentence reads,**

**"The lakes in this study comprise the 117 lakes with the longest (>28 years) and highest temporal resolution time series (greatest number of samples per year)."**

P6L5: ". . . is typically ~3 cm for large lakes. . .". This is a bit overstated. The best case is around 3 cm. Please double check and revise.

**Thanks for this comment. This has been updated to "~5 cm for large lakes" based on the USDA G-REALM website.**

P6L9: ". . .linearly-interpolated each lake's time series. . .". Why not use a model considering annual and semi-annual variations given that you have quite long time series? I suggest using the approach used in previous studies, e.g. (Kleinherenbrink et al., 2015; Villadsen et al., 2014).

**Thanks for this suggestion. In our revision, we have interpolated the time series using ARMA time series modelling with a Kalman filter. The reveiewer is correct that this is a much more robust and accepted way to interpolate the time series.**

P7 2.4: The description of statistical method, BRT should be expanded instead of just saying some R packages are used given that you state this method is novel in your study.

**We are grateful to the reviewer for pointing out this key area for improvement. In our revision we have included the following description of BRTs:**

**"A BRT is an ensemble machine learning approach that differs from conventional statistical techniques which use a single parsimonious model. Instead, BRTs combine the strengths of standard regression trees and boosting—a method for aggregating many models to improve the predictive capacity. The main advantages of BRTs over other statistical models is that they have higher predictive performance, do not require data transformation or outlier elimination, automatically handle complex nonlinear relationships and interactions, and allow for many types of predictor variables and partial missing data."**

P9L26: Lake water level definitely is an integrative metrics of regional water budget. I think this sentence should be rephrased.

**This sentence has been rephrased. Our revised sentence states, "This contrast highlights how global analyses of waterbody surface level variation can enhance our ability to detect hydrological changes."**

P9L26: "significant effects . . . in most lakes", is this overstated?

**Thanks for pointing out our misuse of the term "significant." We have replaced the phrase, "had significant effects on " with the phrase "substantially influenced."**

P11L28-29: As pointed above, I doubt that detrending the PCs helps to separate background climate indices from ongoing climate change. Instead, I would first detrend the time series of temperature, and the perform PCA.

**In our revised manuscript, we have detrended the temperature data prior to the PCA as suggested by the reviewer.**

References

Kleinherenbrink, M., Lindenbergh, R. C. and Ditmar, P. G.: Monitoring of lake level changes on the Tibetan Plateau and Tian Shan by retracking Cryosat SARIn waveforms, J. Hydrol., 521, 119–131, doi:10.1016/j.jhydrol.2014.11.063, 2015.

Villadsen, H., Andersen, O. B. and Stenseng, L.: Annual cycle in lakes and rivers from CryoSat-2 altimetry - The Brahmaputra river, Int. Geosci. Remote Sens. Symp., 894–897, doi:10.1109/IGARSS.2014.6946569, 2014.